# Offline Stochastic Shortest Path: Learning, Evaluation and Towards Optimality

**Ming Yin**[*1,2]        **Wenjing Chen**[*3]        **Mengdi Wang**[4]        **Yu-Xiang Wang**[1]

[1]Department of Computer Science, UC Santa Barbara
[2]Department of Statistics and Applied Probability, UC Santa Barbara
[3]Department of Electrical and Computer Engineering, Texas A&M University
[4]Department of Electrical and Computer Engineering, Princeton University

## Abstract

Goal-oriented Reinforcement Learning, where the agent needs to reach the goal state while simultaneously minimizing the cost, has received significant attention in real-world applications. Its theoretical formulation, *stochastic shortest path* (SSP), has been intensively researched in the online setting. Nevertheless, it remains understudied when such an online interaction is prohibited and only historical data is provided. In this paper, we consider the *offline stochastic shortest path* problem when the state space and the action space are finite. We design the simple *value iteration*-based algorithms for tackling both *offline policy evaluation (OPE)* and *offline policy learning* tasks. Notably, our analysis of these simple algorithms yields strong instance-dependent bounds which can imply worst-case bounds that are near-minimax optimal. We hope our study could help illuminate the fundamental statistical limits of the offline SSP problem and motivate further studies beyond the scope of current consideration.

## 1 INTRODUCTION

Goal-oriented reinforcement learning aims at entering a goal state while minimizing its expected cumulative cost. The interplay between the agent and the environment keeps continuing when the target/goal state is not reached and this causes trajectories to have variable lengths among different trials, which makes it different from (or arguably more challenging than) the finite-horizon RL. In particular, this setting naturally subsumes the *infinite-horizon $\gamma$-discounted* case as one can make up a "ghost" goal state $g$ and set $1 - \gamma$ probability to enter $g$ at each timestep for the latter.

The goal-oriented RL covers many popular reinforcement learning tasks, such as navigation problems (e.g., Mujoco mazes), Atari games (*e.g.* breakout) and Solving Rubik's cube [Akkaya et al., 2019] (also see Figure 1 for more examples). Parallel to its empirical popularity, the theoretical formulation, *stochastic shortest path* (SSP), has been studied from the control perspective (*i.e.* with known transition) since Bertsekas and Tsitsiklis [1991]. Recently, there is a surge of studying SSP from the data-driven aspects (*i.e.* with unknown transition) and existing literatures formulate SSP into the *online reinforcement learning* framework [Tarbouriech et al., 2020, Rosenberg et al., 2020, Cohen et al., 2021, Chen and Luo, 2021, Tarbouriech et al., 2021]. On the other hand, there exists no literature (to the best of our knowledge) formally study the *offline* behavior of stochastic shortest path problem.

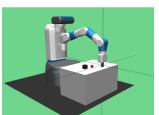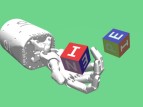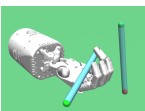

Figure 1: Examples of Goal-oriented RL tasks in OpenAI-Gym environment. The robot can be asked to move-fetch to a position, orient a block or play with a pen.

In this paper, we study the offline counterpart of the stochastic shortest path (SSP) problem. Unlike its online version, we have no access to further explore new strategies (policies) and the data provided are historical trajectories. The goal is to come up with a cost-minimizing policy that can enter the goal state (*policy learning*) or to evaluate the performance of a target policy (*policy evaluation*).

**Why should we study offline SSP?**    Online SSP provides a suitable learning framework for goal-oriented tasks with cheap experiments (*e.g.* Atari games). However, real-world applications usually have high-stake experiments which makes online interactions infeasible. For instance, in the

---

[*]Equal contribution.

*Accepted for the 38th Conference on Uncertainty in Artificial Intelligence* (UAI 2022).

application of logistic transportation, goods need to be delivered to their destinations. How to minimize the transportation cost should be decided/learned beforehand using the logged data. In the aircraft planning, changing flight routes instantaneously could be dangerous and designing routes based on history records is more appropriate for optimizing flying operation budget. In those scenarios, *offline SSP* suffices for treating the practical challenges as it only learns from historical data.

**Our contributions.** In this paper, we provide the first systematic study of the offline stochastic shortest path problem, and consider both *offline policy evaluation* (OPE) and *offline policy learning* tasks. As an initial attempt, we design the simple *value iteration*-based algorithms to tackle the problems and obtain strong statistical guarantees. Concretely, our contributions are four folds.

- For the offline policy evaluation task, we design VI-OPE algorithm (Algorithm 1) under the coverage Assumption 2.4. In particular, our algorithm is *parameter-free* (requires no knowledge about $T^\pi/B^\pi$) and fully executed by the offline data. Theorem 3.1 provides the first statistical guarantee for offline SSP evaluation and nearly matches the statistical efficiency of its finite horizon counterpart (see discussion in Section 3);

- For the offline learning task, we propose *pessimism*-based algorithm PVI-SSP (Algorithm 2) under the Assumption 2.5 and 2.6. Our result (Theorem 4.1) has several merits: it is instance-dependent (as opposed to the worst-case guarantees in the existing online SSP works), enjoys faster $\widetilde{O}(1/n)$ convergence when the system is deterministic, and is also minimax-rate optimal. We believe Theorem 4.1 is (in general) unimprovable for the current tabular setting.

- To understand the statistical limit of offline SSP, we prove the minimax lower bound $\Omega(B_\star\sqrt{\frac{SC^\star}{n}})$ (Theorem 5.1) under the marginal coverage concentrability $\max_{s,a,s\neq g}\frac{d^{\pi^\star}(s,a)}{d^\mu(s,a)} \leq C^\star$. Our Theorem 4.1 can match this rate (up to the logarithmic factor).

- Along the way for solving the problem, we highlight two new technical observations: Lemma 6.1 and Lemma 6.3. The first one depicts the connection between the expected time $T^\pi$ and marginal coverage $d^\pi(s,a)$. As a result, we can express our result without using $T^\pi$ but the ratio-based quantity $\frac{d^\pi(s,a)}{d^\mu(s,a)}$, which matches the flavor of previous finite-horizon RL studies (also see Remark 6.2). The second one is a general dependence improvement lemma that works with arbitrary policy $\pi$ and is the key for guaranteeing minimax optimal rate (also see Remark 6.4). Both Lemmas are general and may be of independent interest.

## 1.1 RELATED WORKS.

Stochastic shortest path itself is a broad topic and we are not aiming for the exhaustive review. Here we discuss two aspects that are most relevant to us.

**Online SSP.** Previous literatures intensively focus on the online aspect of SSP learning. Earlier works consider two types of problems: online shortest path routing problem with deterministic dynamics, which can be solved using the combinatorial bandit technique (*e.g.* György et al. [2007], Talebi et al. [2017]); or SSP with stochastic transitions but adversarial feedbacks [Neu et al., 2012, Zimin and Neu, 2013, Rosenberg and Mansour, 2019, Chen and Luo, 2021, Chen et al., 2021b]. Recently, Tarbouriech et al. [2020] starts investigating general online SSP learning problem and introduce the UC-SSP algorithm to first achieve the no-regret bound $\widetilde{O}(DS\sqrt{ADK})$.[1] Rosenberg et al. [2020] improves this result to $\widetilde{O}(B_\star S\sqrt{AK})$ via a UCRL2-style algorithm with Bernstein-type bonus for exploration. Later, Cohen et al. [2021] eventually achieves the minimax rate $\widetilde{O}(B_\star\sqrt{SAK})$ by a reduction from SSP to finite-horizon MDP. However, the reduction technique requires the knowledge of $B_\star$ and $T_\star$. Most recently, Tarbouriech et al. [2021] proposes EB-SSP which recovers the minimax rate but gets rid of the parameter knowledge (*parameter-free*). When the parameters are known, their results can be *horizon-free*.

Other than the general tabular SSP learning, there are also other threads, *e.g.* Linear MDPs [Min et al., 2021, Vial et al., 2021, Chen et al., 2021a] and posterior sampling [Jafarnia-Jahromi et al., 2021]. Nevertheless, no analysis has been conducted for offline SSP yet.

**Offline tabular RL.** In the offline RL regime, there are fruitful results under different type of assumptions. Yin et al. [2021] first achieves the minimax rate $\widetilde{O}(\sqrt{H^3/nd_m})$ for non-stationary MDP with the strong uniform coverage assumption. Ren et al. [2021] improves the result to $\widetilde{O}(\sqrt{H^2/nd_m})$ for the stationary MDP setting. Later, Rashidinejad et al. [2021], Xie et al. [2021], Li et al. [2022] use the weaker single concentrability assumption and achieve the minimax rate $\widetilde{O}\sqrt{H^3SC^\star/n}$ (or $\widetilde{O}\sqrt{(1-\gamma)^{-3}SC^\star/n}$). Recently, this is further subsumed by the tighter instance-dependent result [Yin and Wang, 2021]. For offline policy evaluation (OPE) task, statistical efficiency has been achieved in tabular [Yin and Wang, 2020], linear [Duan et al., 2020] and differentiable function approximation settings [Zhang et al., 2022].

---

[1] Here the diameter of SSP is defined as $D := \max_{s\in\mathcal{S}} \min_{\pi\in\Pi} T_s^\pi$ and by Lemma 2 of Tarbouriech et al. [2020] $B_\star := \|V^\star\|_\infty \leq c_{\max}D$. In this paper, we consider the dependence on $B_\star$ only since our $c_{\max} = 1$ and this implies $B_\star \leq D$.

## 2 PROBLEM SETUP

**Stochastic Shortest Path.** An SSP problem consists of a *Markov decision process* (MDP) together with an initial state $s_{\text{init}}$ and an extra goal state $g$ and it is denoted by the tuple $M := \langle \mathcal{S}, \mathcal{A}, P, c, s_{\text{init}}, g \rangle$. In particular, we denote $\mathcal{S}' := \mathcal{S} \cup \{g\}$. Each state-action pair $(s, a)$ incurs a bounded random cost (within $[0, 1]$) drawn i.i.d. from a distribution with expectation $c(s, a)$ and will transition to the next state $s' \in \mathcal{S}'$ according to the probability distribution $P(\cdot|s, a)$. Here $\sum_{s' \in \mathcal{S}'} P(s' \mid s, a) = 1$. The goal state $g$ is a termination state with absorbing property and has cost zero (*i.e.* $P(g|g, a) = 1, c(g, a) = 0$ for all $a \in \mathcal{A}$).

The optimal behavior of the agent is characterized by a stationary, deterministic and proper policy that minimizes the expected total cost of reaching the goal state from *any* state $s$. A stationary policy $\pi : \mathcal{S} \to \Delta^{\mathcal{A}}$ is a mapping from state $s$ to a probability distribution over action space $\mathcal{A}$, here $\Delta^{\mathcal{A}}$ is the set of probability distributions over $\mathcal{A}$. The definition of proper policy is defined as follows.

**Definition 2.1** (Proper policies). *A policy $\pi$ is proper if playing $\pi$ reaches the goal state with probability $1$ when starting from any state. A policy is improper if it is not proper. Denote the set of proper policies as $\Pi_{\text{prop}}$.*

**Value and $Q$-functions in SSP.** Any policy $\pi$ induces a *cost-to-go* value function $V^\pi : \mathcal{S} \mapsto [0, \infty]$ defined as

$$V^\pi(s) := \lim_{T \to \infty} \mathbb{E}^\pi \left[ \sum_{t=0}^{T} c(s_t, a_t)|s_0 = s \right], \quad \forall s \in \mathcal{S}$$

and the Q-function is defined as $\forall s, a \in \mathcal{S} \times \mathcal{A}$,

$$Q^\pi(s, a) := \lim_{T \to \infty} \mathbb{E}^\pi \left[ \sum_{t=0}^{T} c(s_t, a_t)|s_0 = s, a_0 = a \right],$$

where the expectation is taking w.r.t. the random trajectory of states generated by executing $\pi$ and transitioning according to $P$. Also, we denote $T_s^\pi := \lim_{T \to \infty} \mathbb{E}[\sum_{t=0}^{T} \mathbf{1}[s_t \neq g]|s_0 = s] = \mathbb{E}[\sum_{t=0}^{\infty} \mathbf{1}[s_t \neq g]|s_0 = s]$ to be the expected time that $\pi$ takes to enter $g$ starting from $s$. By Definition 2.1, $\pi$ is proper if $T_s^\pi < \infty$ for all $s$, and improper if $T_s^\pi = \infty$ for some state $s$. Moreover, by definition it follows $V^\pi(g) = Q^\pi(s, a) = 0$ for all $\pi$ and action $a$. The next proposition is the Bellman equation for the SSP problem.

**Proposition 2.2** (Bellman equations for SSP problem [Bertsekas and Tsitsiklis, 1991]). *Suppose there exists at least one proper policy and that for every improper policy $\pi'$ there exists at least one state $s \in \mathcal{S}$ such that $V^{\pi'}(s) = +\infty$. Then the optimal policy $\pi^\star$ is stationary, deterministic, and proper. Moreover, $V^\star = V^{\pi^\star}$ is the unique solution of the equation $V^\star = \mathcal{L}V^\star$, where*

$$\mathcal{L}V(s) := \min_{a \in \mathcal{A}} \{c(s, a) + P_{s,a}V\} \quad \forall V \in \mathbb{R}^{S'}.$$

*Similarly, for a proper policy $\pi$, $V^\pi$ is the unique solution of $V^\pi = \mathcal{L}^\pi V^\pi$ with $\mathcal{L}^\pi V(s) := \mathbb{E}_{a \sim \pi(\cdot|s)}[c(s, a) + P_{s,a}V]$, $\forall V \in \mathbb{R}^{S'}$. Furthermore, it holds*

$$Q^\star(s, a) = c(s, a) + P_{s,a}V^\star, \quad V^\star(s) = \min_{a \in \mathcal{A}} Q^\star(s, a),$$

$$Q^\pi(s, a) = c(s, a) + P_{s,a}V^\pi, \quad V^\pi(s) = \mathbb{E}_{a \sim \pi(\cdot|s)}[Q^\pi(s, a)]. \tag{1}$$

We use $T_s^\star$ to denote the expected arriving time when coupled with the optimal policy $\pi^\star$ and the proof of Proposition 2.2 can be found in Appendix A.

**The Offline SSP task.** The goal of offline SSP is to reach the goal state but also minimize the cost using offline data $\mathcal{D} := \{(s_0^{(i)}, a_0^{(i)}, c_0^{(i)}, s_1^{(i)}, \dots, s_{T_i}^{(i)})\}_{i=1,\dots,n}$, which is collected by a proper (possibly stochastic) behavior policy $\mu$. The optimal policy is a proper policy $\pi^\star$ (the existence of $\pi^\star$ is guaranteed by the Proposition 2.2) which minimizes the value function for all states, *i.e.*,

$$\pi^\star(s) = \arg \min_{\pi \in \Pi_{\text{prop}}} V^\pi(s). \tag{2}$$

The final learning objective is to come up with a (proper) policy $\widehat{\pi}$ using $\mathcal{D}$ such that the suboptimality gap $V^{\widehat{\pi}}(s_{\text{init}}) - V^\star(s_{\text{init}}) < \epsilon$ for a given accuracy $\epsilon > 0$.

**Some Notations.** In the paper, we may abuse the notation $V^\star$ with $V^{\pi^\star}$, and define $B_\star := \max_s \{V^\star(s)\}$. In addition, we denote $\xi_h^\pi(s, a)$ to be the marginal state-action occupancy at time step $h$ under the policy $\pi$ and $\xi_h^\pi(s)$ the marginal state occupancy at time $h$. Furthermore, we define the *marginal coverage* $d^\pi$ as (given the initial state is $s_{\text{init}}$):

$$d^\pi(s, a) := \sum_{h=0}^{\infty} \xi_h^\pi(s, a), \quad \forall s, a \in \mathcal{S} \times \mathcal{A}. \tag{3}$$

**Remark 2.3.** *The notation of marginal coverage mirrors the marginal state-action occupancy in the infinite horizon $\gamma$-discounted setting but without normalization. Therefore, it is likely that $d^\pi(s, a) > 1$ (or even $\infty$) for the offline SSP problem. Nevertheless, the key Lemma 6.1 guarantees $d^\pi(s, a)$ is finite when $\pi$ is a proper policy. This feature helps formalize the following assumptions in offline SSP.*

### 2.1 ASSUMPTIONS

Offline learning/evaluation in SSP is impossible without assumptions. We now present three required assumptions.

**Assumption 2.4** (offline policy evaluation (OPE)). *We assume both the target policy $\pi$ and behavior policy $\mu$ are proper. In this case, we have $\Pi_{\text{prop}} \neq \emptyset$. Moreover, we assume behavior policy $\mu$ can cover the exploration (state-action) space of $\pi$, i.e. $\forall s, a \in \mathcal{S} \times \mathcal{A}$ s.t. $d_{\bar{s}}^\pi(s, a) := \sum_{h=0}^{\infty} \xi_{h,\bar{s}}^\pi(s, a) > 0$, it implies $d_{\bar{s}}^\mu(s, a) :=$*

$\sum_{h=0}^{\infty} \xi_{h,\bar{s}}^{\mu}(s,a) > 0$, where $d_{\bar{s}}^{\pi}(s,a)$ is the marginal coverage and $\xi_{h,\bar{s}}^{\pi}(s,a)$ the marginal state-action occupancy given the initial state $\bar{s}$. In particular, when $\bar{s} = s_{init}$, we suppress the subscript and use $d^{\pi}, \xi_h^{\pi}$ only.

There are two remarks that are in order.

Assumption 2.4 requires that the behavior policy $\mu$ can explore all the state-action locations that are explored by $\pi$ and this mirrors the necessary OPE assumption made in the standard RL setting (*e.g.* Thomas and Brunskill [2016], Yin and Wang [2020], Uehara et al. [2020]). Otherwise, policy evaluation for SSP would incur constant suboptimality gap even when *infinite many* trajectories are collected.

Moreover, instead of making assumption only on $d_{s_{init}}^{\mu}$, 2.4 assumes $\mu$ can cover $\pi$ when starting from any state $\bar{s}$ (*i.e.* $d_{\bar{s}}^{\pi} > 0$ implies $d_{\bar{s}}^{\mu} > 0$ for all $\bar{s}$). This extra requirement is mild since, by Definition 2.1, a proper policy can reach goal state $g$ with probability 1 when starting from any state $\bar{s}$. Similarly, we need the assumptions for offline learning tasks.

**Assumption 2.5** (offline policy learning). *We assume there exists a deterministic proper policy and the behavior policy $\mu$ is (possible random) proper. Next, by Proposition 2.2, we know there exists a deterministic optimal proper policy $\pi^{\star}$. We assume behavior policy $\mu$ can cover the exploration (state-action) space of $\pi^{\star}$, i.e. $\forall s, a \in \mathcal{S} \times \mathcal{A}$ s.t. $d_{\bar{s}}^{\pi^{\star}}(s,a) := \sum_{h=0}^{\infty} \xi_{h,\bar{s}}^{\pi^{\star}}(s,a) > 0$, it implies $d_{\bar{s}}^{\mu}(s,a) := \sum_{h=0}^{\infty} \xi_{h,\bar{s}}^{\mu}(s,a) > 0$, where $d_{\bar{s}}^{\pi^{\star}}(s,a)$ and $\xi_{h,\bar{s}}^{\pi^{\star}}(s,a)$ is the same notion used in Assumption 2.4. In particular, when $\bar{s} = s_{init}$, we suppress the subscript and use $d^{\pi^{\star}}, \xi_h^{\pi^{\star}}$ only.*

2.5 provides the offline learning version of Assumption 2.4. It echos its offline RL counterpart assumed in Liu et al. [2019], Yin and Wang [2021], Uehara and Sun [2022]. Similar to the offline RL setting (*e.g.* see Yin and Wang [2021] for detailed explanations), this assumption is also required for the tabular offline SSP problem.

**Assumption 2.6** (Positive cost [Rosenberg et al., 2020]). *There exists $c_{\min} > 0$ such that $c(s,a) \geq c_{\min}$ for every $(s,a) \in \mathcal{S} \times \mathcal{A}$.*[2]

This assumption guarantees there is no *"free-cost"* state. With 2.6 it holds that any policy does not reach the goal state has infinite cost, and this certifies the condition in Proposition 2.2 that for every improper policy $\pi'$ there exists at least one state $s$ such that $V^{\pi'}(s) = +\infty$. When $c_{\min}$ is 0, a simple workaround is to solve a perturbed SSP instance with all observed costs clipped to $\epsilon$ if they are below some $\epsilon > 0$, and in this case $c_{\min} = \epsilon > 0$. This will cause only an additive term of order $O(\epsilon)$ (see Tarbouriech et al. [2020]

for online SSP). Therefore, as the first attempt for offline SSP problem, we stick to this assumption throughout the paper. Last but not least, Assumption 2.6 is only used in offline learning problem (Section 4) and our OPE analysis (Section 3) can work well with zero cost.

# 3 OFF-POLICY EVALUATION IN SSP

---

**Algorithm 1** VI-OPE (Value Iteration for OPE problem of Stochastic Shortest Path)

---

1: **Input:** $\epsilon_{\text{OPE}}$, $\mathcal{D} := \{(s_1^{(i)}, a_1^{(i)}, c_1^{(i)}, s_2^{(i)}, \ldots, s_{T_i}^{(i)})\}_{i=1}^n$.

2: **for** $(s,a,s') \in \mathcal{S} \times \mathcal{A} \times \mathcal{S}'$ **do**
3:     Set $n(s,a) = \sum_{i=1}^n \sum_{j=1}^{T_i} \mathbb{I}(s_j^{(i)} = s, a_j^{(i)} = a)$.
4:     **if** $n(s,a) > 0$ **then**
5:        Calculate $\widehat{c}(s,a) = \frac{\sum_{i=1}^n \sum_{j=1}^{T_i} \mathbb{I}(s_j^{(i)}=s, a_j^{(i)}=a) c_j^{(i)}}{n(s,a)}$
6:        $\widehat{P}(s'|s,a) = \frac{\sum_{i=1}^n \sum_{j=1}^{T_i} \mathbb{I}(s_j^{(i)}=s, a_j^{(i)}=a, s_{j+1}^{(i)}=s')}{n(s,a)}$,
7:     **else**
8:        $\widehat{c}(s,a) \leftarrow c_{\min}, \widehat{P}(s'|s,a) \leftarrow \mathbb{I}(s' = g)$.
9:     **end if**
10:    $\diamond$ Perturb the estimated transition kernel
11:    $\widetilde{P}(s'|s,a) = \frac{n(s,a)}{n(s,a)+1} \widehat{P}(s'|s,a) + \frac{\mathbb{I}[s'=g]}{n(s,a)+1}$
12: **end for**
13: $\diamond$ Value Iteration for SSP problem
14: **Initialize:** $V^{(-1)}(\cdot) \leftarrow -\infty$, $V^{(0)}(\cdot) \leftarrow \mathbf{0}$, $i = 0$.
15: **while** $\|V^{(i)} - V^{(i-1)}\|_{\infty} > \epsilon_{\text{OPE}}$ **do**
16:     **for** $(s,a) \in \mathcal{S} \times \mathcal{A}$ **do**
17:        $Q^{(i+1)}(s,a) = \widehat{c}(s,a) + \widetilde{P}_{s,a} V^{(i)}$
18:        $V^{(i+1)}(s) = \langle \pi(\cdot|s), Q^{(i+1)}(s,\cdot) \rangle$
19:        $i \leftarrow i + 1$
20:     **end for**
21: **end while**
22: **Output:** $V^{(i)}(\cdot) \in \mathbb{R}^S$, $V^{(i)}(s_{\text{init}})$.

---

In this section, we assume that Assumption 2.4 holds and consider *offline policy evaluation* (OPE) for the *stochastic shortest path* (SSP) problem. Our algorithmic design follows the natural idea of *approximate value iteration* [Munos, 2005] and is named **VI-OPE** (Algorithm 1). Specifically, VI-OPE approximates (1) by solving the fixed point solution of the empirical Bellman equation associated with estimated cost $\widehat{c}$ and transition $\widetilde{P}$. One highlight is that we construct $\widetilde{P}$ to be the skewed version of the vanilla empirical estimation $\widehat{P}$ by injecting $\frac{1}{n(s,a)+1}$ probability to state $g$ (Line 11 of Algorithm 1).[3] By such a shift, the empirical Bellman operator $\widehat{\mathcal{T}}^{\pi}(\cdot) := \widehat{c}^{\pi} + \widetilde{P}^{\pi}(\cdot)$ becomes a contraction with rate $\rho := \max_{\substack{s,a \\ s \neq g}} \left( \frac{n_{s,a}}{n_{s,a}+1} \right) < 1$ (see Lemma C.1 for details). Hence, *contraction mapping theorem* [Diaz and Margolis, 1968] guarantees the loop (Line 15-21) will end after $O(\log(\epsilon_{\text{OPE}})/\log(\rho))$ iterations for any $\epsilon_{\text{OPE}} > 0$. We

---

[2]Note this assumption only holds for $(s,a) \in \mathcal{S} \times \mathcal{A}$. For goal state $g$, it always has $c(g,a) = 0$ for all $a \in \mathcal{A}$.

[3]This treatment is also used in Tarbouriech et al. [2021].

have the following main result for VI-OPE, whose proof can be found in Appendix F.

**Theorem 3.1** (Offline Policy Evaluation in SSP). *Denote* $d_m := \min\{\sum_{h=0}^{\infty} \xi_h^{\mu}(s,a) : s.t. \sum_{h=0}^{\infty} \xi_h^{\mu}(s,a) > 0\}$, *and* $T_s^{\pi}$ *to be the expected time to hit* $g$ *when starting from* $s$. *Define* $\bar{T}^{\pi} = \max_{\bar{s} \in \mathcal{S}} T_{\bar{s}}^{\pi}$ *and the quantity* $T_{\max} = \max_{i \in [n]} T_i$. *Then when* $n \geq \max\{\frac{49S\iota}{9d_m}, 64(\bar{T}^{\pi})^2 \frac{S\iota}{d_m}, O(\iota/d_m), O(T_{\max}^2 \log(SA/\delta)/d_m^2)\}$, *we have with probability* $1 - \delta$, *the output of Algorithm 1 satisfies* ($\iota = O(\log(SA/\delta))$)

$$
|V^{(i)}(s_{\text{init}}) - V^{\pi}(s_{\text{init}})|
$$
$$
\leq 4 \sum_{s,a,s \neq g} d^{\pi}(s,a) \sqrt{\frac{2\text{Var}_{P_{s,a}}[V^{\pi} + c]\iota}{n \cdot d^{\mu}(s,a)}} + \widetilde{O}(\frac{1}{n}) + \frac{\epsilon_{\text{OPE}}}{1 - \rho}.
$$

*where the* $\widetilde{O}$ *absorbs Polylog term and higher order terms.*

**On statistical efficiency.** First of all, when VI-OPE converges exactly (*i.e.* $\epsilon_{\text{OPE}} = 0$), the output $\widehat{V}^{\pi} := \lim_{i \to \infty} V^{(i)}$ possesses no optimization error (*i.e.* $\epsilon_{\text{OPE}}/(1 - \rho) = 0$) and the (non-squared) statistical rate achieved by VI-OPE is dominated by $O(\sum_{s,a} d^{\pi}(s,a) \sqrt{\frac{\text{Var}_{P_{s,a}}[V^{\pi}+c]\iota}{n \cdot d^{\mu}(s,a)}})$. As a comparison, for the well-studied finite-horizon tabular MDP problem, the statistical limit $O(\sqrt{\sum_{h=1}^{H} \sum_{s,a} d_h^{\pi}(s,a)^2 \frac{\text{Var}_{P_h}[V_{h+1}^{\pi}+c]}{n \cdot d_h^{\mu}(s,a)}})$ has been achieved by Yin and Wang [2020], Duan et al. [2020], Kallus and Uehara [2020] which matches the previous proven lower bound [Jiang and Li, 2016]. Therefore, it is natural to conjecture that the statistical lower bound for SSP-OPE problem has the rate $O(\sqrt{\sum_{s,a} d^{\pi}(s,a)^2 \frac{\text{Var}_{P_{s,a}}[V^{\pi}+c]}{n \cdot d^{\mu}(s,a)}})$. Our simple VI-OPE algorithm nearly matches this conjectured lower bound and only has the expectation outside of the square root. How to obtain the Carmer-Rao-style lower bound for SSP OPE problem and how to close the gap are beyond this initial attempt. We leave these as the future works.

**Parameter-free.** Different from the standard MDPs (*e.g.* finite-horizon, discounted), the SSP formulation generally has variable horizon length which yields no explicit bound on $\|V^{\pi}\|_{\infty}$. Consequently, most of the previous literature that study SSP problem requires the knowledge of expected running time $T^{\pi}/T^{\star}$ or $B^{\pi}/B_{\star}$, the upper bound on $\|V^{\pi}\| / \|V^{\star}\|$ (*e.g.* Tarbouriech et al. [2020], Rosenberg et al. [2020], Cohen et al. [2021], Chen and Luo [2021], Chen et al. [2021a]). In contrast, VI-OPE is fully parameter-free as it requires no prior information about neither $T^{\pi}$ nor $B^{\pi}$ and the main term of our bound does not explicitly scale with those parameters. Last but not least, VI-OPE does not reply on the positive cost Assumption 2.6.

## 4 OFFLINE LEARNING IN SSP

In this section, we consider the offline policy optimization problem. Similar to previous work, we assume the knowledge of an upper bound on the $B_{\star} := \|V^{\star}\|_{\infty}$, which is denoted as $\widetilde{B}$. How to deal with the case when $\widetilde{B}$ is unknown is discussed in Section 7.1. Throughout the section, we suppose Assumption 2.5 and Assumption 2.6 holds.

We introduce our algorithm in Algorithm 2. The main idea behind the algorithm is the pessimistic update of the value function via adding a bonus function to $V^{(i)}$. Here the **bonus function** $b_{s,a}(V) := \sqrt{\frac{2\widehat{c}(s,a)\iota}{n(s,a)}} + \frac{7\iota}{3n(s,a)} + \frac{\widetilde{B}}{n(s,a)} + \frac{16\widetilde{B}\iota}{3n(s,a)} + \max\{2\sqrt{\frac{\text{Var}(\widetilde{P}',V)\iota}{n(s,a)}}, 4\frac{\widetilde{B}\iota}{n(s,a)}\} + 180\sqrt{\frac{3\widetilde{T}\widetilde{B}S}{2n(s,a)n_{\min}}}(\sqrt{\widetilde{B}} + 1)\iota \; \forall(s,a) \in \mathcal{S} \times \mathcal{A}$, where $n_{\max} = \max_{s,a} n(s,a)$ and $n_{\min} = \min_{s,a}\{n(s,a) : n(s,a) > 0\}$. For the goal state $b_{g,a}(V) = 0 \; \forall a \in \mathcal{A}$. Here $\widetilde{T}$ is an upper bound of $T^{\star}$.[4]

---
**Algorithm 2** PVI-SSP (Pessimistic Value Iteration for SSP)
---
1: **Input:** $\epsilon_{\text{OPL}}$, $\mathcal{D} := \{(s_1^{(i)}, a_1^{(i)}, c_1^{(i)}, s_2^{(i)}, \ldots, s_{T_i}^{(i)})\}_{i=1}^n$. $\widetilde{B}$ and $\iota = O(\log(SA/\delta))$. $n_{\max}$ and $b_{s,a}$ see above.
2: **for** $(s, a, s') \in \mathcal{S} \times \mathcal{A} \times \mathcal{S}'$ **do**
3:      Set $n(s,a) = \sum_{i=1}^{n} \sum_{j=1}^{T_i} \mathbb{I}(s_j^{(i)} = s, a_j^{(i)} = a)$.
4:      **if** $n(s,a) > 0$ **then**
5:          Calculate $\widehat{c}(s,a) = \frac{\sum_{i=1}^{n} \sum_{j=1}^{T_i} \mathbb{I}(s_j^{(i)} = s, a_j^{(i)} = a)c_j^{(i)}}{n(s,a)}$
6:          $\widehat{P}(s'|s,a) = \frac{\sum_{i=1}^{n} \sum_{j=1}^{T_i} \mathbb{I}(s_j^{(i)} = s, a_j^{(i)} = a, s_{j+1}^{(i)} = s')}{n(s,a)}$,
7:      **else**
8:          $\widehat{c}(s,a) \leftarrow c_{\min}$, $\widehat{P}(s'|s,a) \leftarrow \mathbb{I}(s' = g)$.
9:      **end if**
10:      $\widetilde{P}'(s'|s,a) = \frac{n_{\max}}{n_{\max}+1}\widehat{P}(s'|s,a) + \frac{\mathbb{I}[s'=g]}{n_{\max}+1}$
11: **end for**
12: ◇ Pessimistic Value Iteration for offline learning
13: **Initialize:** $V^{(-1)}(\cdot) \leftarrow \infty$, $V^{(0)}(\cdot) \leftarrow \widetilde{B} \cdot \mathbf{1}$, $i = 0$.
14: **while** $\|V^{(i)} - V^{(i-1)}\|_{\infty} > 0(\epsilon_{\text{OPL}})$ **do**
15:      **for** $(s,a) \in \mathcal{S}' \times \mathcal{A}$ **do**
16:          $Q^{(i+1)}(s,a) = \min\{\widehat{c}(s,a) + \widetilde{P}'_{s,a}V^{(i)} + b_{s,a}(V^{(i)}), \widetilde{B}\}$
17:          $V^{(i+1)}(s) = \min_a Q^{(i+1)}(s,a)$
18:          $i \leftarrow i + 1$
19:      **end for**
20: **end while**
21: Calculate $\bar{\pi}(\cdot) = \text{argmin}_a Q^{(i)}(\cdot, a)$
22: **Output**: $\bar{\pi}$, $\bar{V}(\cdot) = \min_a Q^{(i)}(\cdot, a)$

---

The use of value iteration to approximate the underlying Bellman optimality equation $V^{\star}(s) = \max_{a \in \mathcal{A}}\{c(s,a) + P_{s,a}V^{\star}\}$, $\forall s \in \mathcal{S}'$ is natural when model components $P, c$ are accurately estimated by $\widetilde{P}', \widehat{c}$. Moreover, comparing to

---
[4]Here we do point the design of $b_{s,a}$ requires $\widetilde{T}$ in addition to $\widetilde{B}$. However, this is not essential as (by Assumption 2.6) $\widetilde{T}$ can be bounded by $\widetilde{B}/c_{\min}$.

VI-OPE, there are several differences for PVI-SSP. First, $\widetilde{P}'$ is chosen according to $n_{\max}$ (instead of $n(s,a)$), which makes $\widetilde{P}'$ "closer" to $\widehat{P}$ but preserves the positive one-step transition to $g$. More importantly, a pessimistic bonus $b_{s,a}$ is added to the value update differently at each state-action location which measures the uncertainty learnt so far from the offline data. Action with higher uncertainty are less likely to be chosen for the next update. Concretely, $\sqrt{\frac{\mathrm{Var}(\widetilde{P}',V^{(i)})}{n}}$ measures the uncertainty of $V^{(i)}$ and $\sqrt{\frac{\widehat{c}}{n}}$ measures the uncertainty of per-step cost $\widehat{c}$.[5] However, to guarantee proper pessimism, we require the knowledge of $\widetilde{B}$ in the design of $b_{s,a}$.

In addition, for analysis purpose we state our result under the regime where the iteration converges exactly and the output $\bar{V}$ (in Line 22) is fixed point of the operator $\widetilde{\mathcal{T}}$ (see Appendix G.2 for details). In practice, one can stop the iteration when the update difference is smaller than $\epsilon_{\mathrm{OPL}}$. We have the following offline learning guarantee for $\bar{\pi}$, which is our major contribution. The proof is deferred to Appendix I.

**Theorem 4.1** (Offline policy learning in SSP). *Denote* $d_m := \min\{\sum_{h=0}^{\infty} \xi_h^{\mu}(s,a) : s.t. \sum_{h=0}^{\infty} \xi_h^{\mu}(s,a) > 0\}$, *and* $T_s^{\pi}$ *to be the expected time to hit $g$ when starting from $s$. Define* $\bar{T}^{\pi} = \max_{\bar{s}\in\mathcal{S}} T_{\bar{s}}^{\pi}$. *Then when $n \geq n_0$, we have with probability $1-\delta$, the output $\bar{\pi}$ of Algorithm 2 is a proper policy and satisfies ($\iota = O(\log(SA/\delta))$)*

$$0 \leq V^{\bar{\pi}}(s_{\mathrm{init}}) - V^{\star}(s_{\mathrm{init}})$$

$$\leq 4 \sum_{s,a,s\neq g} d^{\star}(s,a)\sqrt{\frac{2\mathrm{Var}_{P_{s,a}}[V^{\star}+c]\iota}{n \cdot d^{\mu}(s,a)}} + \widetilde{O}(\frac{1}{n}),$$

*where the quantity* $d_{max} = \max_{s,a} d^{\mu}(s,a)$, *the quantity* $T_{\max} = \max_{i\in[n]} T_i$ *and we define* $n_0 := \max\{\frac{4B_\star - 2c_{min}}{c_{min}d_{max}}, \frac{26^2 \times 2S\iota(\bar{T}^{\star})^2(\sqrt{B_\star}+1)^2}{d_m}, \frac{10^6(\sqrt{\widetilde{B}}+1)^4 S\iota\bar{T}^{\star}\widetilde{T}}{B^\star(\sqrt{B^\star}+1)^2 d_m}$, $O(T_{\max}^2\log(SA/\delta)/d_m^2)\}$.

**On guarantee for policy.** Existing online SSP works measure the algorithm performance using *regret* $R_K^{\mathrm{SSP}} := \sum_{k=1}^{K}\sum_{h=1}^{I^k} c_h^k - K \cdot \min_{\pi\in\Pi_{\mathrm{proper}}} V^{\pi}(s_{\mathrm{init}})$ (*e.g.* [Tarbouriech et al., 2021]) and is different from policy-based regret measurement $R_K := \sum_{k=1}^{K} V_1^{\star}(x_{k,1}) - V_1^{\pi_k}(x_{k,1})$ (*e.g.* Azar et al. [2017]) in standard RL. The notion of $R_K^{\mathrm{SSP}}$ provides the flexibility for policy update even within the episode (since it suffices to minimize $\sum_{h=1}^{I^k} c_h^k$), therefore unable to output a concrete stationary policy for the policy learning purpose. In contrast, Theorem 4.1 provides a policy learning result via bounding the performance of output policy $\bar{\pi}$ explicitly.

**Instance-dependent bound.** Prior online SSP studies focus on deriving better worst-case regret (*e.g.* the minimax rate

---

[5]This is due to $\mathrm{Var}(c) \leq E[c^2] \leq E[c]$ for r.v. $c \in [0,1]$.

is of order $\Theta(B_\star\sqrt{SAK}))$ where the bounds are expressed by the parameters $B_\star/D, S, A$ that lack the characterization of individual instances. In offline SSP, the main term of PVI-SSP is fully expressed by the system quantities with marginal coverage $d^\star$ and $d^\mu$, conditional variance over transition $P$ and cost function $c$. This instance-adaptive result characterizes the hardness of learning better since the magnitude of the bounds changes with the instances. It fully avoids the explicit use of worst-case parameters $B_\star, S, A$.

**Faster convergence.** When the SSP system is deterministic for both cost $c$ and transition $P$, the conditional variances $\mathrm{Var}_{P_{s,a}}[V^\star + c]$ are always zero. In these scenarios, Theorem 4.1 automatically guarantees faster convergence rate $\widetilde{O}(1/n)$ in deterministic SSP learning. Such a feature is not enjoyed by the existing worst-case studies in online SSP as their regrets are dominated by the statistical rate $\widetilde{O}(\sqrt{K})$ even for deterministic systems.

**On optimality.** While instance-dependent, it is still of great interest to understand whether this result is optimal. We provide the affirmative answer by showing a (nearly) matching minimax lower bound under the single concentrability condition in the next section.

# 5  SSP MINIMAX LOWER BOUND

In this section, we study the statistical limit of offline policy learning in SSP. Concretely, we consider the family of problems satisfying bounded partial coverage, *i.e.* $\max_{s,a,s\neq g} \frac{d^{\pi^\star}(s,a)}{d^\mu(s,a)} \leq C^\star$, where $d^\pi(s,a) = \sum_{h=0}^{\infty} \xi_h^\pi(s,a) < \infty$ for all $s,a$ (excluding $g$) for any proper policy $\pi$. This $C^\star$ formally defines the maximum ratio between $\pi^\star$ and $\mu$ in Assumption 2.5. Consequently, we have the following result (the full proof is in Appendix K):

**Theorem 5.1.** *We define the following family of SSPs:*

$$\mathrm{SSP}(C^\star) = \{(s_{\mathrm{init}},\mu,P,c)|\max_{s,a,s\neq g}\frac{d^{\pi^\star}(s,a)}{d^\mu(s,a)} \leq C^\star\},$$

*where* $d^\pi(s,a) = \sum_{h=0}^{\infty} \xi_h^\pi(s,a)$. *Then for any* $C^\star \geq 1$, $\|V^\star\|_\infty = B_\star > 1$, *it holds (for some universal constant $c$)*

$$\inf_{\widehat{\pi}\,\mathrm{proper}}\sup_{(s_{\mathrm{init}},\mu,P,c)\in\mathrm{SSP}(C^\star)} \mathbb{E}_{\mathcal{D}}[V^{\widehat{\pi}}(s_{\mathrm{init}}) - V^\star(s_{\mathrm{init}})]$$

$$\geq c \cdot B_\star\sqrt{\frac{SC^\star}{n}}.$$

Theorem 5.1 reveals for the family with proper policy $\pi^\star$ and $\mu$ with bounded ratio $C^\star$, the minimax lower bound is $\Omega(B_\star\sqrt{\frac{SC^\star}{n}})$. In particular, the dominant term in Theorem 4.1 directly implies this rate (recall $\pi^\star$ is deterministic by 2.5) by the following calculation (assuming $B_\star > 1$ just

like Theorem 5.1):

$$\sum_{s,a,s\neq g} d^\star(s,a)\sqrt{\frac{\mathrm{Var}_{P_{s,a}}[V^\star+c]}{n\cdot d^\mu(s,a)}}$$

$$=\sum_{s,s\neq g} d^\star(s,\pi^\star(s))\sqrt{\frac{\mathrm{Var}_{P_{s,\pi^\star(s)}}[V^\star+c]}{n\cdot d^\mu(s,\pi^\star(s))}}$$

$$\leq\sqrt{\sum_{s,s\neq g}\frac{d^\star(s,\pi^\star(s))}{d^\mu(s,\pi^\star(s))}\cdot\sum_{s,s\neq g}\frac{d^\star(s,\pi^\star(s))\mathrm{Var}_{P_{s,\pi^\star(s)}}[V^\star+c]}{n}}$$

$$\leq\sqrt{\sum_{s,s\neq g} C^\star\cdot\frac{B_\star^2}{n}}=B_\star\sqrt{\frac{SC^\star}{n}}\quad\text{(also see Proposition I.3),}$$

$$(4)$$

where the first inequality uses CS inequality and the second one uses the key Lemma 6.1.[6] This verifies PVI-SSP is near-optimal up to the logarithmic and higher order terms.

# 6 SKETCH OF THE ANALYSIS

In this section, we sketch the proofs of our main theorems. In particular, we focus on describing the procedure of offline policy learning Theorem 4.1. First of all, when the condition $n\geq n_0$ holds, the output $\bar\pi$ is proper with high probability and following this one can conduct standard decomposition:

$$V^{\bar\pi}-V^\star=(V^{\bar\pi}-\bar V)+(\bar V-V^\star)$$

where $V^\star$ is the solution of Bellman optimality operator $\mathcal{T}$ and $\bar V$ is the fixed point solution of the operator $\widetilde{\mathcal{T}}(V)(s)=\min_a\left\{\min\{\widehat c(s,a)+\widetilde P_{s,a}V+b_{s,a}(V),\widetilde B\}\right\}$ (Lem G.6). Also, $V^{\bar\pi}$ satisfies general Bellman equation (Lemma A.1) therefore we first decompose $V^{\bar\pi}-\bar V$ using a *simulation-lemma* style decomposition (Lemma G.8):

$$V^{\bar\pi}-\bar V=\sum_{h=0}^\infty\sum_{\substack{s\\s\neq g}}\xi_h^{\bar\pi}(s)\Big\{(P_{s,\bar\pi(s)}-\widetilde P'_{s,\bar\pi(s)})\bar V$$

$$+c(s,\bar\pi(s))-\hat c(s,\bar\pi(s))-b_{s,\bar\pi(s)}(\bar V)\Big\}$$

By the careful design of $b_{s,a}(\cdot)$, the pessimism guarantees $V^{\bar\pi}-\bar V\leq 0$ (Lemma I.1). For $\bar V-V^\star$, a similar *simulation-lemma* style SSP decomposition (Lemma G.7) follows:

$$\bar V-V^\star\leq\sum_{h=0}^\infty\sum_{\substack{s\\s\neq g}}\xi_h^\star(s)\Big\{(\widetilde P'_{s,\pi^\star(s)}-P_{s,\pi^\star(s)})\bar V$$

$$(5)$$

$$+\hat c(s,\pi^\star(s))-c(s,\pi^\star(s))+b_{s,\pi^\star(s)}(\bar V)\Big\}.$$

Before we proceed to explain about how to bound the residual summations, we present two new lemmas, which help

---

[6]Here since $B_\star>1$, when applying Lemma 6.1, $B_\star$ will dominate $c\in[0,1]$.

characterize the key features of stochastic shortest path problem.

**Lemma 6.1** (Informal version of Lemma D.3). *Let $T^\pi$ be the expected time of arrival to goal state $g$ when applying proper policy $\pi$ and starting from $s_{\text{init}}$, then*

$$T^\pi=\sum_{h=0}^\infty\sum_{\substack{s,a\\s\neq g}}\xi_h^\pi(s,a)=\sum_{\substack{s,a\\s\neq g}}d^\pi(s,a).$$

**Remark 6.2.** *Lemma 6.1 explicitly reflects the connection between the expected arriving time $T^\pi$ and marginal coverage $d^\pi(s,a)$. Unlike the finite-horizon problem where $d_h^\pi$ are probability measures (e.g. see Yin and Wang [2021]), for SSP $d^\pi(s,a)$ can be arbitrary large (for a general policy $\pi$) due to definition 3. Lemma 6.1 guarantees $d^\pi(s,a)<\infty$ for proper policy $\pi$ since by Definition 2.1 $T^\pi<\infty$, and, as a result, make our bound in Theorem 4.1 valid. Note similar result is of less interests in the standard finite-horizon episodic RL since it holds trivially that $H=\sum_{h=1}^H\sum_{s,a}d_h^\pi(s,a)$ and, in SSP, this becomes important as we have undetermined horizon length. With Lemma 6.1, we can get away with estimating the aggregated measure $T^\pi/T^\star$ (like previous online SSP papers did) and use sub-component $d^\pi(s,a)/d^\star(s,a)$ to reflect the behaviors of individual state-action pairs and achieve more instance-dependent results.*

**Lemma 6.3** (Informal version of Lemma D.4). *For any probability transition matrix $P$, policy $\pi$, and any cost function $c\in[0,1]$ associated with $SSP(P,\pi)$. Suppose $V\in\mathbb{R}^{S+1}$ is any value function satisfying order property (where $V(g)=0$), i.e., $V(s)\geq\sum_a\pi(a|s)P_{s,a}V$ for all $s\in\mathcal{S}$, then we have*

$$\sum_{h=0}^\infty\sum_{\substack{s,a\\s\neq g}}\xi_h^\pi(s,a)\mathrm{Var}_{P_{s,a}}(V)\leq 2\|V\|_\infty\cdot V(s_{\text{init}})\leq 2\|V\|_\infty^2.$$

**Remark 6.4.** *Lemma 6.3 can be viewed as a dependence improvement result for SSP problem since it guarantees Theorem 4.1 to achieve the minimax rate via (??). More critically, it widely applies to arbitrary policies assuming the ordering condition holds for $V$. For instance, a direct upper bound using Lemma 6.1 would yield $T^\pi\|V\|_\infty^2$ and $T^\pi$ could be very large or even $\infty$. In contrast, Lemma 6.3 always upper bounds by $2\|V\|_\infty^2$ without extra dependence. Similar result was previously derived in RL, e.g. Lemma 3.4 of Yin and Wang [2020] and also Ren et al. [2021], but their result only applies to $V^\pi$ due the analysis via law of total variances and ours applies to all $V$ (satisfying ordering condition) through only the telescoping sum.*

Now we go back to bounding (5). First of all, by leveraging Lemma 6.1, we are able to bound the $\infty$-norm of $\bar V-V^\star$ as (see Theorem H.1)

$$\|\bar V-V^\star\|_\infty\leq 30\sqrt{\frac{\bar T^\star B_\star\iota}{nd_m}}(\sqrt{B_\star}+1)\qquad(6)$$

which is a crude/suboptimal bound that serves as an intermediate step for the final bound.

**What give rise to instance-dependencies.** Next, we apply *empirical Bernstein inequality* for structure $(\widetilde{P}'_{s,\pi^\star(s)} - P_{s,\pi^\star(s)})\bar{V}$ and $\widehat{c}(s,\pi^\star(s)) - c(s,\pi^\star(s))$ separately. In particular, since both $\bar{V}$ and $\widetilde{P}'_{s,\pi^\star(s)}$ depend on data, therefore Bernstein concentration cannot be directly applied. Informally, we can surpass this hurdle by decomposing

$$(\widetilde{P}' - P)\bar{V} = (\widetilde{P}' - P)(\bar{V} - V^\star) + (\widetilde{P}' - P)V^\star.$$

In this scenario, concentration can be readily applied to $(\widetilde{P}' - P)V^\star$ and crude bound (6) is leveraged here for bounding $(\widetilde{P}' - P)(\bar{V} - V^\star) \leq \left\|\widetilde{P}' - P\right\|_1 \left\|\bar{V} - V^\star\right\|_\infty$. As explained by Zanette and Brunskill [2019], the use of Bernstein concentration is the key for characterizing the structure of problem instance via the expression of conditional variance $\text{Var}_{P_{s,a}}(V^\star)$.

**On the proof for VI-OPE.** At a high level, the proof for VI-OPE (Theorem 3.1) shares the same flavor as that of Theorem 4.1. Ideally, in finite horizon setting the tighter analysis could be conducted by following the pipeline of Section B.7 in Duan et al. [2020], where the dominant error of $\widehat{V}^\pi - V^\pi$ (where $\widehat{V}^\pi = \lim_{i\to\infty} V^{(i)}$ in Algorithm 1) can be decomposed as:

$$\frac{1}{n}\sum_{i=1}^n \sum_{h=0}^\infty \frac{\xi_h^\pi(s_h^{(i)}, a_h^{(i)})}{\xi_h^\mu(s_h^{(i)}, a_h^{(i)})}(Q^\pi - (c + V^\pi))(s_h^{(i)}, a_h^{(i)})$$

Applying Freedman's inequality for the above martingale structure, one can hope for a tighter rate $O(\sqrt{\sum_{s,a} d^\pi(s,a)^2 \frac{\text{Var}_{P_{s,a}}[V^\pi + c]}{n \cdot d^\mu(s,a)}})$. However, such a procedure will have technical issue for SSP problem since: (1) SSP has stationary transition $P$ and $n(s,a)$ is computed via collecting all the transitions that encounter $s,a$ for tighter dependence. This breaks the sequential ordering that is needed for martingale.[7] (2) Even if we have a martingale, the martingale difference will incorporate an infinite sum that could be arbitrary large. Both facts indicate Freedman's inequality cannot be directly applied due to the technical hurdle.

Lastly, the lower bound proof uses a generalized Fano's argument (Lemma L.5), followed by reducing estimation problem to testing. The packing set of hard MDP instances is based on the modification of Rashidinejad et al. [2021] so that Gilbert-Varshamov Lemma L.1 can be applied.

# 7 DISCUSSIONS

---

[7]Note Duan et al. [2020] Corollary 1 considers time-inhomogeneous MDP and each $P_t$ can be estimated stage-wisely so the decomposition forms a martingale.

## 7.1 THE KNOWLEDGE OF $B_\star/\widetilde{B}$

While VI-OPE (Algorithm 1) is parameter-free, our policy learning algorithm PVI-SSP (Algorithm 2) requires $\widetilde{B}$ in the pessimistic bonus design. Since $\widetilde{B}$ is an upper bound of $B_\star$, one natural idea is to use VI-OPE to provide an upper bound estimation given that a proper policy is provided. This idea is summarized as below.

**Proposition 7.1** (Alternative offline learning algorithm VI-OPE+PVI-SSP). *Suppose we are provided with an arbitrary proper policy $\pi$ (e.g. some previously deployed strategy). In this scenario, one can equally halve the data $\mathcal{D}$ into $\mathcal{D}_1$ and $\mathcal{D}_2$, and use $\mathcal{D}_1$ to evaluate $V^\pi$. The $\infty$-norm of VI-OPE output serves as surrogate for $\widetilde{B}$ and uses as an input for computing $b_{s,a}$. Next, use $\mathcal{D}_2$ to run PVI-SSP (with calculated $b_{s,a}$).*

The above procedure will not deteriorate the theoretical guarantee since $\widetilde{B}$ is only used in $O(1/n)$ terms and the estimation error can only be higher order terms. This means we will end up with the same dominant term as Theorem 4.1.

## 7.2 ON HIGHER ORDER TERMS.

In our analysis of Theorem 4.1, while the dominant $\widetilde{O}(\sqrt{1/n})$ term is near-optimal, the higher order term $\widetilde{O}(1/n)$ is not and depends on the parameters including $\widetilde{T}$, $\widetilde{B}$ and $d_m$ (e.g. check the last line of (54)). In particular, if one can remove the polynomial dependence of $\widetilde{T}$, then the result is called *horizon-free* [Tarbouriech et al., 2021]. One potential approach for addressing the higher order dependence could be the recent development of robust estimation in RL [Wagenmaker et al., 2021]. As the initial attempt for offline SSP, this is beyond our scope and we leave it as the future work.

## 7.3 FUTURE DIRECTIONS

**SSP under weaker conditions.** Following previous works, we consider stochastic shortest path problem with a discrete action space $\mathcal{A}$ and non-negative cost bounded by $c \in [0,1]$. However, the convergence of SSP can hold under much weaker conditions. For instance, Bertsekas and Yu [2013] shows under *compactness and continuity condition*, *i.e.* for each state $s$ the admissible action set $\mathcal{A}(s)$ is a compact metric space and a subset of $\mathcal{A}$ where (for all $s'$) transition $P(s'|s,\cdot)$ are continuous functions over $\mathcal{A}(s)$ and the cost function $c(s,\cdot)$ is lower semi-continuous over $\mathcal{A}(s)$, value iteration/policy iteration will still work under mild assumptions. This extends our setting (*e.g.* cost can even be negative) and how to conduct SSP learning in this case remains open.

**Extension to linear MDP case.** Another natural and promising generalization of the current study is the offline linear

MDP for SSP problem. In the study of offline RL with linear MDPs, Jin et al. [2021] shows the provable efficiency, Zanette et al. [2021] improves the result in the *linear Bellman complete* setting and Yin et al. [2022] leverages variance-reweighting for least square objective to obtain the near-optimal result. Adopting their useful results in offline SSP problem is hopeful.

# 8 CONCLUSION

In this paper, we initiate the study of *offline stochastic shortest path* problem. We consider both *offline policy evaluation* (OPE) and *offline policy learning* tasks and propose the simple value-iteration-based algorithms (VI-OPE and PVI-SSP) that yield strong theoretical guarantees for both evaluation and learning tasks. To complement the discussion, we also provide an information-theoretical lower bound and it certifies PVI-SSP is minimax rate optimal. We hope our work can draw further attention for studying offline SSP setting.

### Acknowledgements

Ming Yin and Yu-Xiang Wang are partially supported by NSF Awards #2007117 and #2003257. MY would like to thank Tongzheng Ren for helpful discussions.

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
