# OpenReview forum: "Offline Stochastic Shortest Path: Learning, Evaluation and Towards Optimality"
_auai.org/UAI/2022/Conference — UAI 2022 Poster_

### Official Review · Reviewer_1TrE · 2022-04-08

**Q2(1) Originality/Novelty:** 3
**Q2(2) Significance/Impact:** 3
**Q2(3) Correctness/Technical Quality:** 3
**Q2(6) Clarity Of Writing:** 2
**Q6 Overall Score:** 6
**Q8 Confidence In Your Score:** 2

**Q1 Summary And Contributions:**

The paper looks at the offline version of the stochastic shortest path problem. SSP is representative for goal-oriented reinforcement learning having a wide range of applications. The offline version of SSP is applicable to real-world situations where online interactions are infeasible or too expensive. The main contribution consists of offline policy evaluation as well as offline policy learning algorithms with strong statistical guarantees.


**Q2 Assessment Of The Paper:**

More detailed information regarding each of these aspects is given below:

**Q2(4) Quality Of Experiments (Optional):**

1: Poor: The experimental evaluation is flawed or the results fail to adequately support the main claims.

**Q2(5) Reproducibility:**

2: Fair: Key resources (e.g., proofs, code, data) are unavailable but key details (e.g., proof sketches, experimental setup) are sufficiently well-described for an expert to confidently reproduce the main results.

**Q3 Main Strengths:**

The main strength of the paper is an extensive theoretical analysis of offline policy evaluation as well as offline policy learning algorithms for the offline SPP problem.


**Q4 Main Weakness:**

The presentation is quite dense using a relatively heavy notation which makes it inaccessible to the wider AI community. It is actually quite difficult to follow the technical details without some supporting running examples.


**Q5 Detailed Comments To The Authors:**

I was actually quite surprised  that the paper doesn't include an empirical evaluation. It is typically the case these days to evaluate reinforcement learning algorithms on standard benchmarks in order to provide some evidence on their performance in practice.

The offline SSP problem appears to be new and therefore I think some sort of evaluation would strengthen the paper considerably.


**Q7 Justification For Your Score:**

The paper provides a strong theoretical contribution but lacks an empirical evaluation of the proposed algorithms.


**Q9 Complying With Reviewing Instructions:**

1: Yes.

---

### Official Review · Reviewer_6Tyj · 2022-04-12

**Q2(1) Originality/Novelty:** 3
**Q2(2) Significance/Impact:** 3
**Q2(3) Correctness/Technical Quality:** 3
**Q2(6) Clarity Of Writing:** 3
**Q6 Overall Score:** 7
**Q8 Confidence In Your Score:** 2

**Q1 Summary And Contributions:**

The paper studies the problem of learning policies to minimize the cost of reaching a specific state in an MDP in an offline setting. This means that a policy is learned based on historical data and not by interacting with the environment. This is motivated by real-world situations which involve the control of systems where the cost of making a mistake is high. The authors present algorithms for policy learning and evaluation and prove instance-based bounds on their performance.

**Q2 Assessment Of The Paper:**

More detailed information regarding each of these aspects is given below:

**Q2(5) Reproducibility:**

3: Good: Key resources (e.g., proofs, code, data) are available and key details (e.g., proofs, experimental setup) are sufficiently well-described for competent researchers to confidently reproduce the main results.

**Q3 Main Strengths:**

The problem tackled by the authors appears to be well-motivated, important and intuitive. The problem formulation appears to be original, though I am not very confident in those assessments as this is far from my research area. The paper is also quite clear at the high level, even though the content is very technical. The authors do a good job structuring their results in a way that makes them easy to understand and not too hard to verify, though due to time constraints I had to skip verifying some of the lemmas.

**Q4 Main Weakness:**

While the overall structure of the paper is well laid out, some paragraphs are confusing as the authors try to condense too much information in too little space. The authors cite a lot of previous work, but it is not clear which of those papers are most relevant to the present work.

**Q5 Detailed Comments To The Authors:**

- p. 5, left middle - reply-> rely

last sentences in Remark 6.2 unclear

since both V̄ and P e s,π
? (s) depend on data, therefore
Bernstein concentration cannot be directly applied - very confusing

**Q7 Justification For Your Score:**

The paper addresses an important problem and delivers a clear result. I am not an expert in the field, however, so I do not have a lot of confidence in my assessment.

**Q9 Complying With Reviewing Instructions:**

1: Yes.

---

### Official Review · Reviewer_4ZVs · 2022-04-13

**Q2(1) Originality/Novelty:** 3
**Q2(2) Significance/Impact:** 2
**Q2(3) Correctness/Technical Quality:** 3
**Q2(6) Clarity Of Writing:** 2
**Q6 Overall Score:** 5
**Q8 Confidence In Your Score:** 3

**Q1 Summary And Contributions:**

The work investigates an offline version of the standard stochastic shortest path (SSP) problem, giving two algorithms for it and analyzing their properties.


**Q2 Assessment Of The Paper:**

More detailed information regarding each of these aspects is given below:

**Q2(5) Reproducibility:**

3: Good: Key resources (e.g., proofs, code, data) are available and key details (e.g., proofs, experimental setup) are sufficiently well-described for competent researchers to confidently reproduce the main results.

**Q3 Main Strengths:**

This is a solid technical study of offline SSP and two of its algorithms.


**Q4 Main Weakness:**

Presentation of the work is quite abstract, leaving much of the background unexplained.

An experimental study would strengthen the work.


**Q5 Detailed Comments To The Authors:**

It would be good to clarify (in the introduction) the differences between "offline shortest path" and the two components (I) constructing a conventional MDP/SSP from historical data, and (II) applying conventional (non-learning) MDP/SSP algorithms. Is this (only?) about the scarceness of the historical data, which makes it difficult/impossible to do (I) well?

There are other readability issues. E.g. the "data" in "The Offline SSP task" seems to be a collection of sequences of state, action, cost, but it would be good to confirm this assumption.

Bibliography: Two dots missing in the "o" in "Gyorgy" and ' in the lower case "a" in "Andras".

Bibliography should also otherwise be cleaned up. E.g. things that should be upper case is in lower case, including "RL", "Markov"


**Q7 Justification For Your Score:**


The paper is very technical, a bit exhausting, and would significantly benefit from improved presentation as well as better placement in the context provided by earlier works.


**Q9 Complying With Reviewing Instructions:**

1: Yes.

---

### Decision · Program_Chairs · 2022-05-15

**Decision:**

Accept (Poster)

**Comment:**

Meta Review: Ultimately this paper ended up just above the threshold of UAI acceptance--the authors gave a very nice response.  In the end, two reviewers requested experimental validation of the result to provide evidence of practical performance of the proposed algorithm -- this would strengthen the paper since constants can matter for practical implementation.  As a minor remark, I note that some of the motivating examples (Mujoco) with seemingly continuous state and action spaces can be misleading if the authors require finite state and action spaces; and returning to the experimental concern, it's always nice to evaluate on the problems that one motivates with.